# A Novel *NKX2-5* Variant in a Child with Left Ventricular Noncompaction, Atrial Septal Defect, Atrioventricular Conduction Disorder, and Syncope

**DOI:** 10.3390/jcm11113171

**Published:** 2022-06-02

**Authors:** Yuya Yamada, Kazushi Yasuda, Yukiko Hata, Naoki Nishida, Keiichi Hirono

**Affiliations:** 1Department of Cardiology, Aichi Children’s Health and Medical Center, Obu 474-8710, Japan; y.yamada19880823@gmail.com (Y.Y.); kazyasuda@vs01.vaio.ne.jp (K.Y.); 2Department of Legal Medicine, Faculty of Medicine, University of Toyama, Toyama 930-0194, Japan; yhatalm@med.u-toyama.ac.jp (Y.H.); nishida0717@yahoo.co.jp (N.N.); 3Department of Pediatrics, Faculty of Medicine, University of Toyama, Toyama 930-0194, Japan

**Keywords:** left ventricular noncompaction, atrial septal defect, conduction disorder, syncope

## Abstract

The *NKX2-5* gene encodes a transcription factor and is actively involved in heart formation and development. A pediatric case with its variant and left ventricular noncompaction (LVNC) has not been reported. A 12-year-old girl with a history of a surgery for atrial septal detect was referred because of syncope during exercise. The electrocardiogram showed atrioventricular block, and the echocardiogram revealed prominent trabeculations in the left ventricular wall, suggesting LVNC. A novel heterozygous variant in the *NKX2-5* gene (NM_004387.1: c.255_256delCT, p.Phe86fs) was identified. *NKX2-5* variants should be considered in cases with LVNC, congenital heart disease, arrhythmia, and syncope to prevent sudden cardiac death.

## 1. Introduction

The NK2 homeobox 5 (*NKX2-5*) gene is located on chromosome 5, comprising two exons and encoding 324 amino acids. Its variants have been associated with atrial septal defects (ASD), dilated cardiomyopathies, and atrioventricular (AV) conduction disorders, with variable penetrance and expressiveness [1]. Left ventricular noncompaction cardiomyopathy (LVNC) is characterized by prominent ventricular trabeculations on cardiovascular imaging [2]. However, there are few case reports of *NKX2-5* variants with LVNC, ASD, AV conduction disorder, and syncope in adults.

Herein, we present a pediatric patient with a novel *NKX2-5* variant and phenotype overlap between these multiple cardiac involvements.

## 2. Case Presentation

A 12-year-old girl with a history of a surgery for secundum-type ASD (a diameter of 20 mm) at 4 years old was referred to our hospital because of syncope. The situation of the syncope was that when she was playing soccer, she fainted soon after feeling something hit her chest strongly, and recovered consciousness without resuscitation. She had no family history of cardiomyopathy, ASD, or sudden cardiac death (SCD). No other metabolic abnormalities were observed.

The electrocardiogram showed first- and second-degree AV block (PR interval 0.244 s) (Figure 1). Chest X-ray imaging did not detect cardiomegaly and pulmonary congestion. The treadmill test showed a decrease in ST during exercise, and the exercise testing with myocardial scintigraphy showed a localized decrease in perfusion in the left ventricular anterior and posterior wall and apex. The cardiac catheterization and angiography showed normal cardiac function without any coronary artery involvement, and the acetylcholine spasm provocation test was negative (cardiac index = 3.48 L/min/m^2^). The repeated echocardiogram revealed prominent trabeculations in the left ventricular lateral and posterior wall and apex (ratio of noncompacted layer to compacted layer at end-systole and end-diastole > 2.0 at each segment), suggesting LVNC, confirmed with cardiac magnetic resonance imaging (Figure 1). LV ejection fraction was 64%, and enlargement of LV and late gadolinium enhancement were not observed.

At age 14, genetic testing using next-generation sequencing with a cardiomyopathy-associated gene panel, which included 182 genes, was performed after obtaining informed consent from the patient’s parents Appendix A. A novel heterozygous *NKX2-5* variant (i.e., deletion variant (NM_004387.1: c.255_256delCT, p.Phe86fs)) was identified (Figure 1). This variant has not been described previously in public databases of the general population and was indicated as deleterious by multiple bioinformatic predictors Appendix A. The patient’s parents were healthy and did not carry the same variant, suggesting a de novo variant and supporting evidence of its probable pathogenicity.

After genetic counseling, the patient quit exercising, including soccer. She is now 16 years old and has not had any subsequent major cardiac events.

## 3. Discussion

To the best of our knowledge, this is the first pediatric case of LVNC, ASD, AV conduction disorders, and syncope caused by a novel variant in the *NKX2-5* gene.

A pathogenic genetic variant in *NKX2-5* was first found to be associated with congenital heart disease, such as ASD [3]. In patients with LVNC, various forms of *NKX2-5* variant have been identified (Table 1). In this case, nonsense-mediated mRNA decay caused by the deletion variant may lead to haplo-insufficiency. There does not appear to be a genotype–phenotype correlation with respect to structural defects, but all reported missense variants are located in the *NKX2-5* homeobox (amino acids 138–197). The pathogenicity of the present variant showed uncertain significance according to the guidelines of the American College of Medical Genetics and Association for Molecular Pathology. However, the clinical manifestation of the present variant was similar to previous reports (Table 1). First, in previous reports, patients with variants in the *NKX2-5* gene had LVNC, ASD, and conduction disorders, in addition to a family history of sudden death. Second, the 2-base deletion in exon 2 in *NKX2-5* is predicted to produce a stop codon at residue 86. Other variants from previous reports were located in more 3′ ends of exon 2. For these reasons, we believe that the present novel variant was pathogenic for this patient.

*NKX2-5* was found to be linked to AV conduction disorders because of its role in the regulation of electrophysiological properties, and AV block is the most commonly reported phenotype [4]. Recent studies reported that a murine model of *NKX2-5* knockout and knock-in (Arg52Gly, Nkx2-5+/R52G) showed PR prolongation and AV block. The sizes of AV node of these mice were small, suggesting atrophy of the AV conduction axis during development because of insufficiency of *NKX2-5* [5,6]. SCD is generally attributed to progressive AV block in patients with *NKX2-5* variants. A previous study reported that 44% of families with SCD had *NKX2-5* variants combined with cardiomyopathy [7]. The patient experienced syncope. Apart from vasovagal syncope, the event occurred during exercise. The presence of atrioventricular block and the variant in the NKX2-5 gene may have been influenced by arrhythmogenic factors, and SCD requires careful attention in the future.

In LVNC, the most common genes associated with arrhythmia are *HCN4* and *SCN5A*, and their variants are important in cases with sick sinus syndrome and syncope rather than *NKX2-5* [2]. The clinical difference between *NKX2-5* and these genes is that *NKX2-5* is often associated with congenital heart disease. The identification of variants in the context of LVNC may be useful in diagnosing cases with *NKX2-5* variants associated with these diverse symptoms.

**Table 1 jcm-11-03171-t001:** Previous cases with variants in NXKX2-5 gene.

Author	Year	Age	Sex	Codon	Protein	Variant Type	Coding Effect	CM	CHD	Arrhythmia	Syncope	FHx of CM	FHx of CHD	FHx of SCD	FHx of Arrhythmia
Bermudez-Jimenez [8]	2017	48	F	c.499G > A	p.Glu167Lys	substitution	missense	LVNC	ASD	CAVB, NSVT	no	yes	yes	no	yes
Morlanes-Gracia [9]	2021	42	M	c.542A > C	p.Gln181Pro	substitution	missense	LVNC	-	CAVB, NSVT	no	yes	yes	yes	yes
Doza [10]	2018	30	F	c.549G > C	p.Lys183Asn	substitution	missense	LVNC	ASD	IAVB	no	yes	yes	yes	yes
Present case	2022	12	F	c.255_256delCT	p.Phe86fs	deletion	frameshift	LVNC	ASD	IAVB	yes	no	no	no	no
Ouyang [11]	2011	adult	M	c.510_511dup	p.Leu171Argfs*6	insertion	frameshift	LVNC	ASD	IAVB	yes	no	yes	yes	yes
Ross [12]	2020	36	F	c.677_680del	p.Asp226Alafs*5	deletion	frameshift	LVNC	ASD	NSVT	no	yes	no	yes	no
Guntheroth [4]	2012	19	M	c.783del	p.Ala262Argfs*32	deletion	frameshift	LVNC	-	2:1AVB	no	yes	yes	no	yes
Ross [12]	2020	34	F	c.744C > A	p.Tyr248Ter	substitution	nonsense	LVNC	-	IAVB, VT	no	no	no	no	no

CM: cardiomyopathy; CHD: congenital heart disease; FHx: family history; SCD: sudden cardiac death; LVNC, left ventricular noncompaction; ASD: atrial septal defect; CAVB: complete atrioventricular block; NSVT: non-sustained ventricular tachycardia; IAVB: first degree atrioventricular block; VT: ventricular tachycardia.

## 4. Conclusions

Genetic testing should be considered in a case with LVNC, ASD, AV block, and syncope. The identification of *NKX2-5* variants may have important diagnostic and management implications to prevent SCD.

## Figures and Tables

**Figure 1 jcm-11-03171-f001:**
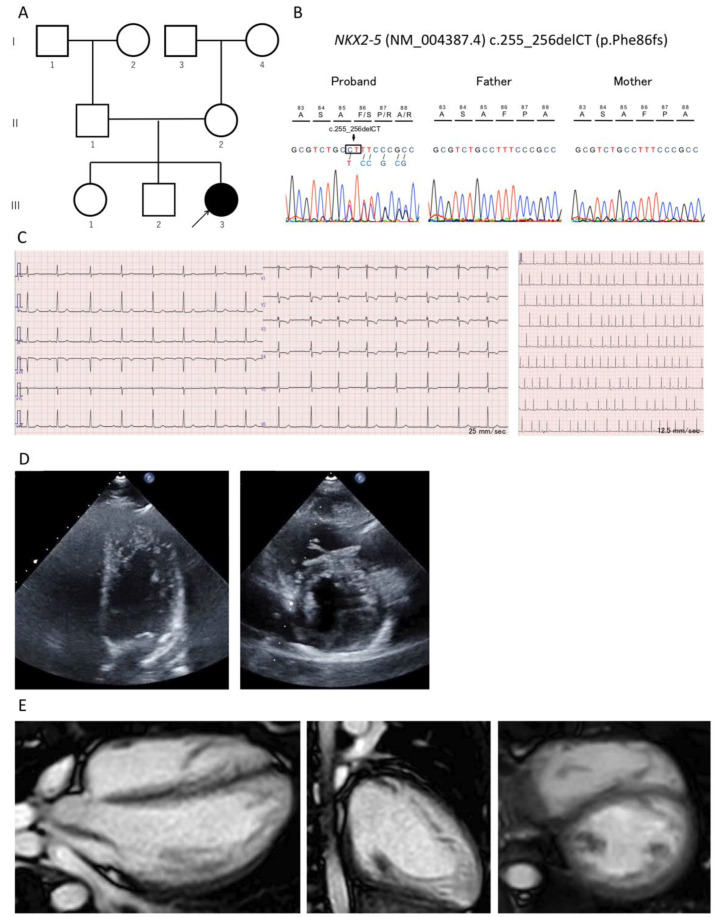
Result of the genetic testing and multiple images of left ventricular noncompaction. (**A**) Family pedigree. (**B**) Results of the Sanger sequence of target alleles. (**C**) Electrocardiograms showing first-degree (**left**) and second-degree (**right**) atrioventricular block. (**D**) Ultrasound images showing an abnormal, highly trabeculated left ventricular myocardium: four-chamber view (**left**) and short-axis view (**right**). (**E**) Cardiac magnetic resonance imaging showing an abnormal, highly trabeculated left ventricular myocardium.

## Data Availability

The authors confirm that the data supporting the findings of this study are available within the article and its Appendix A.

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
