# Peer review of "A Novel NKX2-5 Variant in a Child with Left Ventricular Noncompaction, Atrial Septal Defect, Atrioventricular Conduction Disorder, and Syncope"

_jcm, 2022, doi:10.3390/jcm11113171_

Round 1

Reviewer 1 Report

In this manuscript authors report an interesting case of a NKX2-5 gene mutation. The case is well presented, nevertheless there are some pitfalls.

Case presantation- Comments

  1. How was the second degree AV block (with syncope) was managed?  It was permanent or transient?
  2. Myocardial scintigraphy was abnormal. Which is the underlying mechanism for this finding?
  3. Did CMR revealed late gadolenium enhancement, which has been associated with increased risk for SCD?
  4. Why genetical testing was performed 2 years after the index event? Please explain.

Discussion

  1. Are the electophysiological properties regulated by the NKX2-5 known? Please provide more information.
  2. According to the authors, which are the clinical findings that should alert cardiologists to proceed with genetical testing in these cases?The coexistence of CHD and LVNC alone are enough? 
  3. How should the asymptomatic carriers of this mutation should be managed according ti the literature and the authors?

Author Response

Reviewer 1

We thank the associated editor for his/her careful reading. We have addressed the reviewer’s questions, revising our manuscript as follows. We believe that these amendments have made our manuscript more informative for the readers.

Case presentation- Comments

  1. How was the second-degree AV block (with syncope) was managed? It was permanent or transient?

>> We performed Holter monitoring twice during observation period, and second-degree AV block was occasionally observed and pause was not observed. Therefore, the patient did not receive any treatment.

  1. Myocardial scintigraphy was abnormal. Which is the underlying mechanism for this finding?

>> The acetylcholine spasm provocation test was negative and coronary stenosis was not observed on catheterization. LVNC often present abnormal findings on myocardial scintigraphy which were indicative of various extents of myocardial damage. It was speculated that the underlying mechanism might be failure of the coronary microcirculation associated with abnormal arrest of compacted process.

According to your suggestion, we modified the sentence as follows in case report.

“The cardiac catheterization and angiography showed normal cardiac function without any coronary artery involvement and the acetylcholine spasm provocation test was negative (cardiac index=3.48 L/min/m²).”

Reference:

Gao XJ, Li Y, Kang LM, Zhang J, Lu MJ, Wan JY, Luo XL, He ZX, Zhao SH, Yang MF, Yang YJ. Abnormalities of myocardial perfusion and glucose metabolism in patients with isolated left ventricular non-compaction. J Nucl Cardiol. 2014;21:633-42.

  1. Did CMR revealed late gadolinium enhancement, which has been associated with increased risk for SCD?

>> Late gadolinium enhancement was not observed on CMR.

According to your suggestion, we modified the following sentence as follows in case report.

LV ejection fraction was 64%, and enlargement of LV and late gadolinium enhancement were not observed.

  1. Why genetical testing was performed 2 years after the index event? Please explain.

>> At the beginning of the visit, coronary artery disease was suspected. However, cardiomyopathy was emerged after repeated modality examinations including echocardiography, CMR, and CT. As a result, it took 2 years to perform genetic testing.

Discussion

  1. Are the electophysiological properties regulated by the NKX2-5 known? Please provide more information.

>> Thank you for suggesting an important point. Recent studies reported that NKX2-5 knockout and knock-in (Arg52Gly, Nkx2-5+/R52G) murine model showed PR prolongation and AV block. The sizes of AV node were smaller than control, suggesting atrophy of AV conduction axis during development because of insufficiency of NKX2-5. According to your suggestion, we added the following sentence in discussion.

“Recent studies reported that murine model of NKX2-5 knockout and knock-in (Arg52Gly, Nkx2-5+/R52G) showed PR prolongation and AV block. The sizes of AV node of these mice were small, suggesting atrophy of AV conduction axis during development because of insufficiency of NKX2-5.”

References:

Briggs LE, Takeda M, Cuadra AE et al. Perinatal loss of Nkx2-5 results in rapid conduction and contraction defects. Circ Res 2008;103:580-90.

Chowdhury R, Ashraf H, Melanson M et al. Mouse Model of Human Congenital Heart Disease: Progressive Atrioventricular Block Induced by a Heterozygous Nkx2-5 Homeodomain Missense Mutation. Circ Arrhythm Electrophysiol 2015;8:1255-64.

  1. According to the authors, which are the clinical findings that should alert cardiologists to proceed with genetical testing in these cases? The coexistence of CHD and LVNC alone are enough?

>> Thank you for suggesting an important point. As noted in the last paragraph of discussion, we think that genetic testing should be considered when coexistence of CHD, LVNC, arrhythmia, and syncope. According to your suggestion, we added CHD and arrhythmia in the last sentence of abstract.

  1. How should the asymptomatic carriers of this mutation should be managed according the literature and the authors?

>> Thank you for suggesting an important point. This disease has been considered an adult disease. However, we proved that this disease even occurs in childhood. Therefore, we think that it is important to screen cardiac defects in the asymptomatic carriers soon after a variant in NKX2-5 gene is identified in a proband and follow up properly even if the asymptomatic carriers do not have any symptoms.

Thank you for your kindly constructive comments and your consideration of the revised version.

Sincerely,

Reviewer 2 Report

The authors propose a new genetic marker in a patient for LV non compaction with known atrial defect.  The proposed gene NKX2-5 with c.255_256delCT as a new candiate marker for LVNC as de novo mutation is interesting. This gene is associated with  atrial septal defect with atrioventricular conduction defect and also tetralogy of Fallot. There are several other authors that have identified mutations in NK homeobox associated with LVNC such as Pauli et al. (1999) who described a distal 5q deletion in a 7 y.o girl with LVNC  and septal atrial defect. The authors identify a new such case which may be in fact part of a genetic syndrome with predominat cardiac abnormalities. I would comment on the fact if the authors can provide some additional data regarding the size of the atrial defect, if there were any other metabolic derangements such as serum sodium or potasium etc and if there was any Holter monitoring for incidental arrhythmias.

Author Response

Reviewer 2

We thank the associated editor for his/her careful reading. We have addressed the reviewer’s questions, revising our manuscript as follows. We believe that these amendments have made our manuscript more informative for the readers.

The authors propose a new genetic marker in a patient for LV non compaction with known atrial defect. The proposed gene NKX2-5 with c.255_256delCT as a new candidate marker for LVNC as de novo mutation is interesting. This gene is associated with atrial septal defect with atrioventricular conduction defect and also tetralogy of Fallot.

There are several other authors that have identified mutations in NK homeobox associated with LVNC such as Pauli et al. (1999) who described a distal 5q deletion in a 7-year-old girl with LVNC and septal atrial defect. The authors identify a new such case which may be in fact part of a genetic syndrome with predominant cardiac abnormalities.

>> Thank you for your valuable comment. However, the case report that you suggested (Pauli et al. 1999) was large deletion in chromosome 5 whereas our case report was only 2 codon deletions in NKX2-5 gene. Considering this aspect, we addressed that our case report was a first pediatric case with LVNC, congenital heart disease, arrhythmia, and syncope during exercise.

I would comment on the fact if the authors can provide some additional data regarding the size of the atrial defect, if there were any other metabolic derangements such as serum sodium or potassium etc and if there was any Holter monitoring for incidental arrhythmias.

>> The ASD diameter was 20mm. We added a diameter in the manuscript.

>> No other metabolic abnormalities were observed. We added it in the manuscript.

>> We performed Holter monitoring twice, and second-degree AV block was occasionally observed and pause was not observed.

Thank you for your kindly constructive comments and your consideration of the revised version.

Sincerely,